# GABins: Gated Attention Bins for Depth Estimation
## Conference Submissions

## Abstract

Estimating pinpoint-accuracy depth from a single RGB image is challenging because it is an ill-posed problem as infinitely many 3D scenes can be projected to the same 2D scene. We propose a method to convert monocular depth estimation from a regression problem to a classification problem. The natural long-range property of the attention mechanism allows it to make good use of the global effective information in the fine features. We compute gated attention on highest-resolution feature map to obtain the association information between features to divide the depth range into bins whose center value is estimated adaptively per image. The final depth values are estimated by linearly combining with the results of the multiscale feature fusion and bin centers. We call our new building block GABins. Experiments demonstrate that our method improves the performance across all metrics on both the KITTI and NYUv2 datasets compared to previous methods. Further, we explore the model generalization capability via zero-shot testing.

## 1 Introduction

Monocular depth estimation is one of the most important basic modules in 3D vision, and it is critical and necessary to accurately and quickly acquire depth information in the application scenarios in industrial applications such as autonomous driving, human-computer interaction, virtual reality and robotics.At the same time, it is easier to distinguish the boundaries of objects by obtaining the before-and-after relationship from the depth map, which simplifies the algorithms for CV tasks such as 3D target detection and segmentation, scene understanding.However, monocular depth estimation is an ill-posed problem, as the same 2D image may correspond to countless 3D scenes, so it is necessary to analyze and constrain the local and global information in the 2D image in order to obtain an accurate depth value.

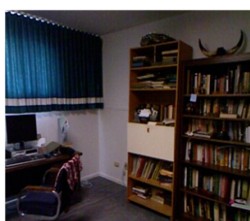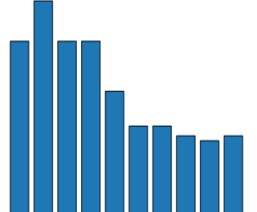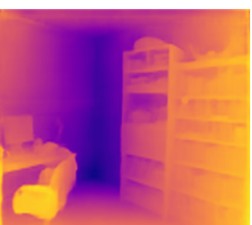

Figure 1: Illustration of GABins. **Left:** input RGB image. **Middle:** histogram of the GABins predicted depth-bin-centers. **Right:** depth predicted by our model.

Eigen et al. Eigen et al. (2014) first utilized CNN to predict depth from a single image by integrating global and local information. In our opinion, subsequent approaches to improve the accuracy of monocular depth estimation using neural network models are divided into three main categories.The first type of method is to increase the training data Chen et al. (2020); Ranftl et al. (2020); Bhat et al. (2023), mixing images of multiple scenes as the training dataset to improve the generalization ability

of the model.The second class of methods is based on the traditional method of depth estimation Hua & Tian (2016); Ricci et al. (2018); Yuan et al. (2022), integrating Markov random fields or conditional random fields into deep networks to improve the accuracy of the model.The third class of methods is based on the geometric properties that depth maps have and use these known geometric rules to guide model training and constrain the model Yin et al. (2019; 2023); Patil et al. (2022). We also observe that several works formulate regression problems as classification tasks to improve performance Fu et al. (2018); Bhat et al. (2021; 2022).It has been demonstrated that regression lags in its ability to learn high-entropy features, which in turn leads to the lower mutual information between the learned representation and the target output.

In this paper, we propose a method to divide the depth distribution (as shown in Figure 1 ) according to the image gated attention features,which convert the monocular depth estimation from a regression problem to a classification problem, and then combine the results of multi-scale feature fusionto fully analyze and fuse the image information for depth estimation.The depth discretization module adjusts the distribution of adaptive depth values for each input image. The characteristics of the distribution of objects on different images vary greatly, for example, indoor images with the walls have their depth distribution range confined to a very small area, while outdoor images with the sky have their depth distribution range extending to a great distance.The natural long-range property of the attention mechanism allows it to make good use of the global effective information in the fine features. For the highest-resolution feature map output by the encoder, we compute gated attention on this feature map to obtain the association information between features.The network based on the correlation between pixel features, can better focus on areas with sufficient information. In these areas, the distribution intervals of depth values are small and the classification is dense. Gated attention Hua et al. (2022) has quadratic complexity, but compared to the standard transformer, it has faster speed, lower memory usage and better results.

Monocular depth estimation differs from other vision tasks that focus more on features at a certain level. Depth estimation is not only to predict the overall image extent, but also to predict the scene and position changes in the image. So the features of image level, region level and pixel level should be utilized with sufficient attention. The spatial resolution of the output features form usual backbone network gradually decreases from large to small, where the low resolution corresponding to semantically expressive features, which cannot capture more detailed information while maintaining a high spatial sensitivity.High semantic features do not provide enough position information, and large scale feature maps have detailed and powerful localization information. Fusing feature maps with different spatial resolutions and combining large and small sensory fields can make full use of multiple scales of features.We apply this idea in depth prediction to obtain more feature information through multi-scale fusion.

Our method can reach the latest state-of-the-art level on the indoor dataset NYU and the outdoor dataset KITTI by experiments. Our main contributions are the following:

• We design a module GABins which capture gated attention in the low-level features and divide the predicted depth range into bins where the bin widths change per image. The final depth values are estimated by linearly combining with the results of the multiscale feature fusion and bin centers.

• We propose to use a single-head gated attention unit other than multi-head self-attention to reduce computational complexity, and the Laplace normalization function to make the model train more stable.

•We show a improvement for supervised monocular depth estimation on some metrics for the two most popular indoor and outdoor datasets, NYUv2 and KITTI.

## 2 RELATED WORK

**Monocular depth estimation.** Monocular depth estimation has always been a challenging task,due to the rapid development of deep networks combined with the traditional monocular depth estimation methods Saxena et al. (2008); Wang et al. (2015), researchers have proposed many ingenious architectures Saxena et al. (2005); Bhat et al. (2021); Fan et al. (2017); Yin et al. (2019); Huynh et al. (2020) to constantly improve the accuracy of monocular depth estimation.Some researchers propose to discretize continuous depth into a number of intervals and cast the depth network learning as an ordinal regression problem. Fu et al. (2018) propose to perform the discretization using a

spacing-increasing discretization strategy instead of the uniform discretization strategy. Bhat et al. (2021) propose to divide the predicted depth range into bins where the bin widths change per image.

**Gated attention.** Gated attention can be seen as a layer that introduces a gating mechanism to alleviate the burden of self-attention. Attention provides the key mechanism that captures contextual information from the entire scope by modeling pairwise interactions. The transformer architecture built by self-attention and position-wise feed-forward has stood out for its impressive empirical success on a wide range of language and vision tasks Dosovitskiy et al. (2010); Touvron et al. (2021). Shazeer et al. Shazeer (2020) proposed additional variations on the transformer FFN layer which use GLU or one of its variants in place of the first linear transformation and the activation function. Hua et al. Hua et al. (2022) proposed to add attention to GLU, called GAU. As compared to transformer layers, each GAU layer is cheaper. More importantly, its quality relies less on the precision of attention, which allows the use of a weaker single-head attention with minimal quality loss. In this paper, we use GAU for the feature map with the highest resolution (the lowest-level) for attention operation. Moreover, a single-head GAU is used to make the model simpler and light-weighted. The Laplace normalization function is used to make the model training effective and stable.

**Multi-scale fusion.** Multi-scale fusion is mostly used in the computer vision task.By using different ways of utilizing features with different spatial resolutions, fusing feature maps with different spatial resolutions and combining large and small perceptual fields, we can make full use of multiple scales of features and have better detection results for both large and small objects.

# 3 METHOD

In this section, we describe the proposed monocular depth estimation network with a novel GABins.

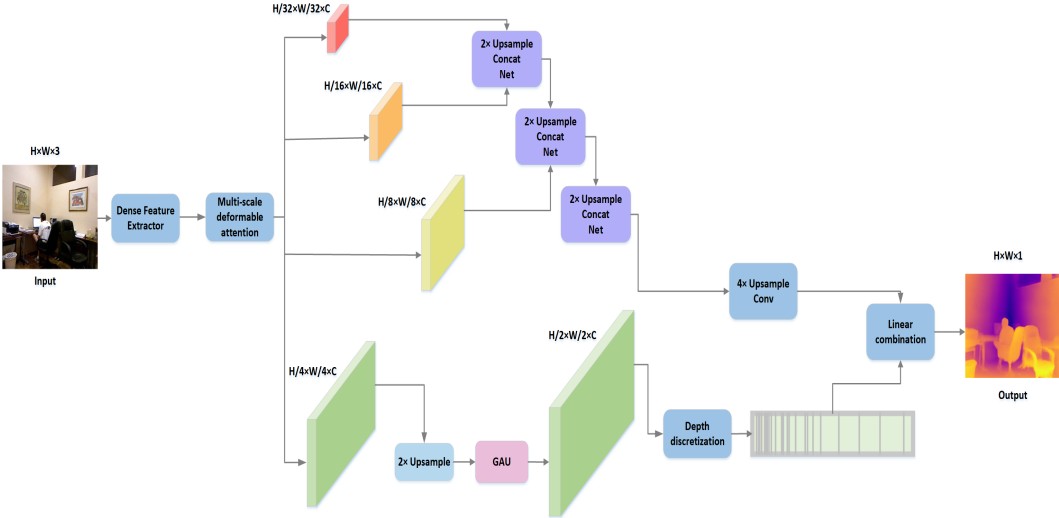

Figure 2: Overview of the proposed network architecture. The network is composed of dense feature extractor, GABins.

## 3.1 NETWORK ARCHITECTURE

As seen from Figure 2 , we follow an encoding-decoding scheme that obtains hierarchical feature maps and then recovers back to the original image resolution ($H \times W$) for dense prediction. Due to memory limitation of current GPU hardware, we use $H/2 \times W/2$ to facilitate better learning with larger batch sizes. The final depth map is computed by simply bilinearly up-sampling to $H \times W$.

Because our method requires different processing for feature maps with different spatial resolutions, the backbone should be able to output features with different spatial resolutions, with typical networks such as ResNet He et al. (2016) and its derivative networks. Swin-transformer Liu et al.

(2021) not only introduces windows multi-head self-attention and shifted windows multi-head self-attention to reduce the computational complexity and enables it to process high-resolution images. In addition, it uses patch merging to implement a hierarchical structure so that it can output hierarchical feature maps. Swin-transformer is used as a general-purpose backbone for computer vision and achieved state-of-the-art performance on various vision tasks such as for monocular depth estimation.

The backbone is swin-transformer which is used as a dense feature extractor. It produces hierarchical feature maps from $H/4 \times W/4 \times C$ to $H/32 \times W/32 \times 8C$. First, all features are passed through the multi-scale deformable attention module, the number of channels is reduced and unified as $C$ from $8C$, $4C$, and $2C$. Secondly, for the lowest-level features, they are up-sampled from $H/4 \times W/4 \times C$ to $H/2 \times W/2 \times C$ for GAU. The GAU is used to extract attention in the fine features from a global field. Then, these features are discretized by the discretization module to discretize the depth range and obtain the depth width bins and the center value of the bins. Thirdly, we fuse the features obtained from the first step to get a $H/4 \times W/4 \times C$ feature map. Finally, the fused features and the center value are linearly combined to output a depth map ($H \times W$).

### 3.2 SINGLE-HEAD GATED ATTENTION

We use GAU Hua et al. (2022) to calculate attention for the lowest-level feature. The lowest-level feature has the highest resolution output from the encoder, which contains a lot of detailed information. By performing attention calculation on this feature map, the relationship among features can be learned. It enhances the acquisition of global information for estimating the depth value. Single-head gated attention has been empirically shown as performant as vanilla multi-head attention Vaswani et al. (2017), but it is more concise. Unlike the standard transformer is built by the self-attention layer and Feed-Forward Networks (FFN), GAU is a new design that integrates the two layers into one. The standard FFN is a two-layer MLP model:

$$O = \phi \left( X W_u \right) W_o \tag{1}$$

where $X \in R^{n \times d}$: input, $W_u \in R^{d \times e}$: MLP layer, $W_o \in R^{e \times d}$: MLP layer, $\phi$: activation function, $O \in R^{n \times d}$: output.

In Gehring et al. (2017); Shazeer (2020), it is found that the FFN layer using gated linear unit (GLU) works better, as follows:

$$O = (U \odot V)W_o, \;\; U = \phi_u \left( X W_u \right), \;\; V = \phi_v \left( X W_v \right) \tag{2}$$

where $W_u, W_v \in R^{d \times e}$: MLP layer, $\odot$: element-wise multiplication.

Since each row of the matrix $U$ and $V$ is operated independently and there is no interaction between the tokens in the FFN, so FFN cannot replace attention. In order to make up for this deficiency, it is to add the connection between tokens to $U$ and $V$:

$$O = (U \odot AV)W_o \tag{3}$$

where $A \in R^{n \times n}$ is attention matrix, which is responsible for fusing the information between tokens. In this way, $O$ contains the interactions between tokens so that it acts as the attention. If $A$ is a unitary matrix I, then it is an FFN of the GLU. If $A$ is an all-one matrix, then it is an ordinary attention matrix.

GLU's dependence on attention is not as strong as the standard transformer Vaswani et al. (2017); Devlin et al. (2018), and $A$ can be simplified according to the standard scaled-dot self-attention:

$$A = \frac{1}{n} relu^2 \left( \frac{\mathcal{Q}\left( Z \right) \mathcal{K}(Z)^T}{\sqrt{s}} \right), Z = \phi_z \left( X W_z \right) \tag{4}$$

where $W_z \in R^{d \times s}$; $s$ is the head size of the attention; $\mathcal{Q}$, $\mathcal{K}$ are affine transformation; $relu^2$ is the square of $relu$. David et al. So et al. (2021) recently introduced the squared of relu function via architecture search techniques, which has shown faster convergence speed and competitive generalization performance on language tasks Hua et al. (2022). However, one issue of the squared of relu function is that neither its range nor its gradient is bounded, leading to unstable model training. To

address this issue, Ma et al. Ma et al. (2022) proposed an attention function based on the Laplace function:

$$A = \frac{1}{2n} \times \left[ 1 + erf(\frac{\mathcal{Q}(Z)\mathcal{K}(Z)^T}{\sqrt{s}} - \mu/\sigma\sqrt{2}) \right] \tag{5}$$

where $erf(\bullet)$ is the error function; $\mu$ and $\sigma$ are two coefficients of the above function to approximate the square of relu function. It is experimentally demonstrated that the model has better accuracy and stability when $\mu = \sqrt{1/2}$ and $\sigma = \sqrt{1/4\pi}$.

### 3.3 MULTI-SCALE FEATURES FUSION

The encoder generates multi-level features with high-resolution fine-grained and low-resolution coarse ones. By aggregating the information from different layers, the fused multi-scale features can combine both local and global information. There are three steps in the fusion process. The first step is to align the resolution of coarse feature up-sampled with the resolution of the fine feature. The second step is to concatenate the resolution increased coarse feature in the first step with fine feature in the channel dimension. The third step is to pass the concatenated feature through a two-layer convolution network. Repeat the above process to complete the fusion for the three different scale features.

### 3.4 DEPTH DISCRETIZATION

Features output by the gated attention module, following the Bhat et al. (2021) practice, we use an MLP head over the first output embedding. The MLP head uses a ReLU activation and outputs an N-dimensional vector $b'$. Finally, we normalize the vector $b'$ such that it sums up to 1, to obtain the bin-widths vector $b$ as follows:

$$b_i = \frac{b'_i + \varepsilon}{\sum_{j=1}^{N}(b'_j + \varepsilon)} \tag{6}$$

where $\varepsilon = 10^{-3}$. The samll positive $\varepsilon$ ensures each bin-width is strictly positive. Then calculate the depth-bin-center:

$$c(b_i) = d_{\min} + (d_{\max} - d_{\min})(b_i/2 + \sum_{j=1}^{i-1} b_j) \tag{7}$$

Features obtained after multi-scale fusion are passed through a $1 \times 1$ convolutional layer to obtain N-channels which is followed by a Softmax activation as Softmax scores $p_k, k = 1, ..., N$, for each pixel. Finally, at each pixel, the final depth value $\tilde{d}$ is calculated from the linear combination of Softmax scores at that pixel and the depth-bin-centers $c(b)$ as follows:

$$\tilde{d} = \sum_{k=1}^{N} c(b_k)p_k \tag{8}$$

### 3.5 LOSS FUNCTION

Following previous works Eigen et al. (2014); Lee et al. (2019); Bhat et al. (2021); Lee et al. (2021), we use a Scale-Invariant Logarithmic loss proposed by Eigen et al. (2014) to supervise the training. Given the ground-truth depth map, we first calculate the logarithm difference between the predicted depth map and the real depth:

$$g_i = \log \tilde{d}_i - \log d_i \tag{9}$$

where $d_i$ is the ground-truth depth value and $\tilde{d}_i$ is the predicted depth at pixel $i$.

Then for $T$ pixels which have ground-truth in an image, the scale-invariant loss is computed as

$$\mathcal{L}_{pixel} = \alpha \sqrt{\frac{1}{T} \sum_i g_i^2 - \frac{\lambda}{T^2}(\sum_i g_i)^2} \tag{10}$$

where $\lambda$ is a variance minimizing factor, and $\alpha$ is a scale constant. In our experiments, $\lambda$ is set to 0.85 and $\alpha$ is set to 10 following previous works [4].

## 4 EXPERIMENTS

We conduct an extensive set of experiments on the standard depth estimation from a single image datasets for both indoor and outdoor scenes. In the following, we first briefly describe the datasets and the evaluation metrics, and then present quantitative comparisons to the state-of-the-art in supervised monocular depth estimation.

### 4.1 DATASETS

**NYU Depth V2.** NYU Depth V2 dataset contains 120K RGB and depth pairs with size of $480 \times 640$, acquired as video sequences using a Microsoft Kinect from 464 indoor scenes. We follow the official train/test split as previous works, using 249 scenes for training and 215 scenes (654 images) for testing. From the total 120K image-depth pairs, because of asynchronous capturing rates between RGB images and depth maps, we associate and sample them using timestamps by even-spacing in time, resulting in 24231 image-depth pairs for the training set. Using raw depth images and camera projections provided by the dataset, we align the image-depth pairs for accurate pixel registration. At testing, we compute the final output by taking the average of an image's prediction and the prediction of its mirror image which is commonly used in previous work.

**KITTI.** KITTI is the most commonly used benchmark with outdoor scenes that provides stereo images and corresponding 3D laser scans of outdoor scenes captured using equipment mounted on a moving vehicle. It contains categories of "city", "residential", "road", and "campus". The RGB images have resolution of around $1241 \times 376$ while the corresponding depth maps are of very low density with lots of missing data. We train our network on a subset of around 26K images, from the left view, corresponding to scenes not included in the 697 images test set. We train our network on a random crop of size $1120 \times 352$.

### 4.2 IMPLEMENTATION DETAILS

Our work is implemented in Pytorch Paszke et al. (2019) and experimented on Nvidia GeForce RTX 2080 Ti GPUs. For training, we use the AdamW optimizer Loshchilov & Hutter (2017) with weight-decay $10^{-2}$. We use the 1-cycle policy Smith & Topin (2019) for the learning rate with $max\_lr = 3.5 \times 10^{-4}$, linear warm-up from $max\_lr/25$ to $max\_lr$ for the first 30% of iterations followed by cosine annealing to $max\_lr/75$. Total number of epochs is set to 30 with batch size 4.

Table 1: Comparison of performances on the NYU-Depth-v2 dataset. The reported numbers are from the corresponding original papers. Best results are in bold, second best are underlined. "*" means using additional data for training.

| Method | $\delta_1 \uparrow$ | $\delta_2 \uparrow$ | $\delta_3 \uparrow$ | Abs_REL $\downarrow$ | RMSE $\downarrow$ | $\log_{10} \downarrow$ |
|---|---|---|---|---|---|---|
| DORN Fu et al. (2018) | 0.828 | 0.965 | 0.992 | 0.115 | 0.509 | 0.051 |
| VNL Yin et al. (2019) | 0.875 | 0.976 | 0.994 | 0.108 | 0.416 | 0.048 |
| BTS Lee et al. (2019) | 0.885 | 0.978 | 0.994 | 0.110 | 0.392 | 0.047 |
| DAVHuynh et al. (2020) | 0.882 | 0.980 | 0.996 | 0.108 | 0.412 | - |
| PackNet-SAN* Guizilini et al. (2021) | 0.892 | 0.979 | 0.995 | 0.106 | 0.393 | - |
| AdaBins Bhat et al. (2021) | 0.903 | 0.984 | 0.997 | 0.103 | 0.364 | 0.044 |
| DPT* Ranftl et al. (2021) | 0.904 | 0.988 | 0.998 | 0.110 | 0.357 | 0.045 |
| PWALee et al. (2021) | 0.892 | 0.985 | 0.997 | 0.105 | 0.374 | 0.045 |
| P3Depth Patil et al. (2022) | 0.830 | 0.971 | 0.995 | 0.130 | 0.450 | 0.056 |
| LocalBins Bhat et al. (2022) | 0.907 | 0.987 | 0.998 | 0.099 | 0.357 | 0.042 |
| NeWCRFs Yuan et al. (2022) | 0.922 | 0.992 | 0.998 | 0.095 | 0.334 | 0.041 |
| iDisc Piccinelli et al. (2023) | **0.940** | **0.993** | **0.999** | **0.086** | **0.313** | **0.037** |
| Ours | 0.935 | 0.992 | 0.998 | **0.086** | 0.314 | 0.039 |

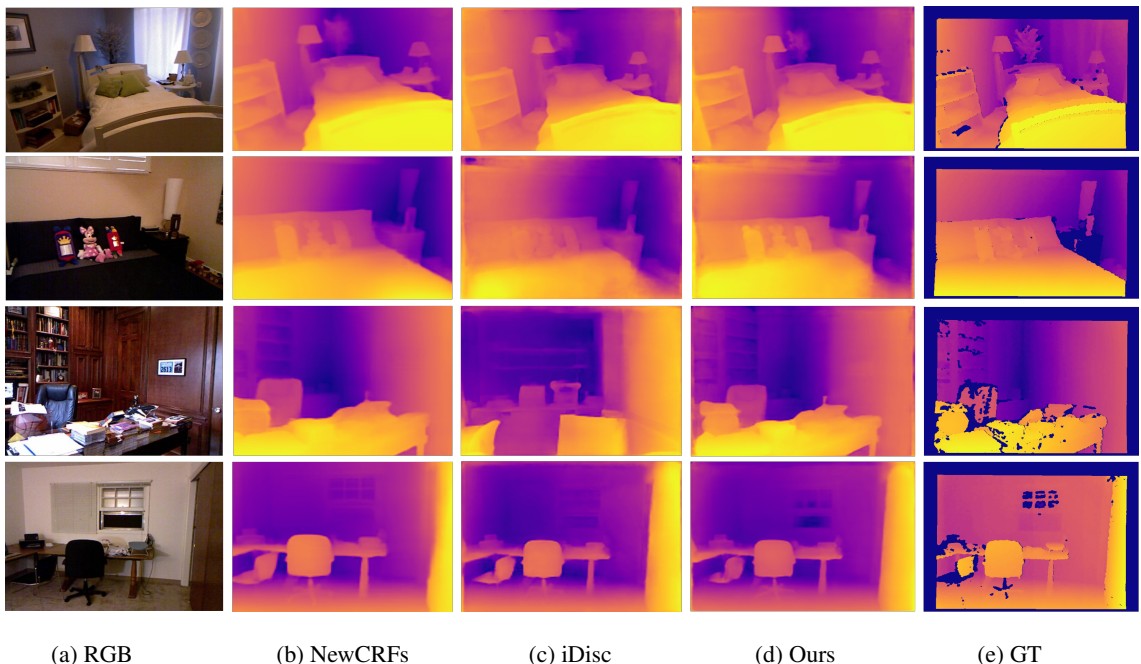

| (a) RGB | (b) NewCRFs | (c) iDisc | (d) Ours | (e) GT |
| --- | --- | --- | --- | --- |

Figure 3: Qualitative comparison with the state-of-the-art on the NYU-Depth-v2 dataset.

Table 2: Comparison of performances on the KITTI dataset. We compare our network against the state-of-the-art on this dataset. The reported numbers are from the corresponding original papers. Measurements are made for the depth range form 0m to 80m. Best results are in bold, second best are underlined. "*" means using additional data for training.

| Method | $\delta_1 \uparrow$ | $\delta_2 \uparrow$ | $\delta_3 \uparrow$ | Abs_REL $\downarrow$ | RMSE $\downarrow$ | RMSE_log $\downarrow$ | Sq_Rel $\downarrow$ |
| --- | --- | --- | --- | --- | --- | --- | --- |
| DORN Fu et al. (2018) | 0.932 | 0.984 | 0.995 | 0.072 | 2.727 | 0.120 | 0.307 |
| VNLYin et al. (2019) | 0.938 | 0.990 | 0.998 | 0.072 | 3.258 | 0.117 | - |
| BTS Lee et al. (2019) | 0.956 | 0.993 | 0.998 | 0.059 | 2.756 | 0.096 | 0.241 |
| PackNet-SAN Guizilini et al. (2021) | 0.955 | - | - | 0.062 | 2.888 | - | - |
| AdaBins Bhat et al. (2021) | 0.964 | 0.995 | 0.999 | 0.058 | 2.360 | 0.088 | 0.190 |
| DPT* Ranftl et al. (2021) | 0.959 | 0.995 | 0.999 | 0.062 | 2.573 | 0.092 | - |
| PWA Lee et al. (2021) | 0.958 | 0.994 | 0.999 | 0.060 | 2.604 | 0.093 | 0.221 |
| P3Depth Patil et al. (2022) | 0.959 | 0.994 | 0.999 | 0.060 | 2.519 | 0.095 | 0.206 |
| NeWCRFs Yuan et al. (2022) | 0.974 | 0.997 | 0.999 | 0.052 | 2.129 | 0.079 | 0.155 |
| iDisc Piccinelli et al. (2023) | **0.977** | **0.997** | **0.999** | **0.050** | **2.067** | **0.077** | **0.145** |
| Ours | 0.975 | **0.997** | **0.999** | 0.051 | 2.167 | 0.079 | 0.151 |

### 4.3 EVALUATION METRICS

We use the standard six metrics used in prior work Eigen et al. (2014) to compare our method against state-of-the-art. These error metrics are defined as: average relative error (REL): $\frac{1}{n} \sum_p^n \frac{|y_p - \hat{y}_p|}{y}$; root mean squared error (RMSE): $\sqrt{\frac{1}{n} \sum_p^n (y_p - \hat{y}_p)^2}$; average (log10) error: $\frac{1}{n} \sum_p^n |log_{10}(y_p) - log_{10}(\hat{y}_p)|$; threshold accuracy($\delta_i$): % of $y_p$, s.t. $\max\left(\frac{y_p}{\hat{y}_p}, \frac{\hat{y}_p}{y_p}\right) = \delta < thr$, for $thr = 1.25, 1.25^2, 1.25^3$; where $y_p$ is a pixel in depth image $y$, $\hat{y}_p$ is a pixel in the predicted depth image $\hat{y}$, and $n$ is the total number of pixels for each depth image.

### 4.4 COMPARISON WITH STATE-OF-THE-ART

In this section, we detail the comparison of our methods with state-of-the-art methods.

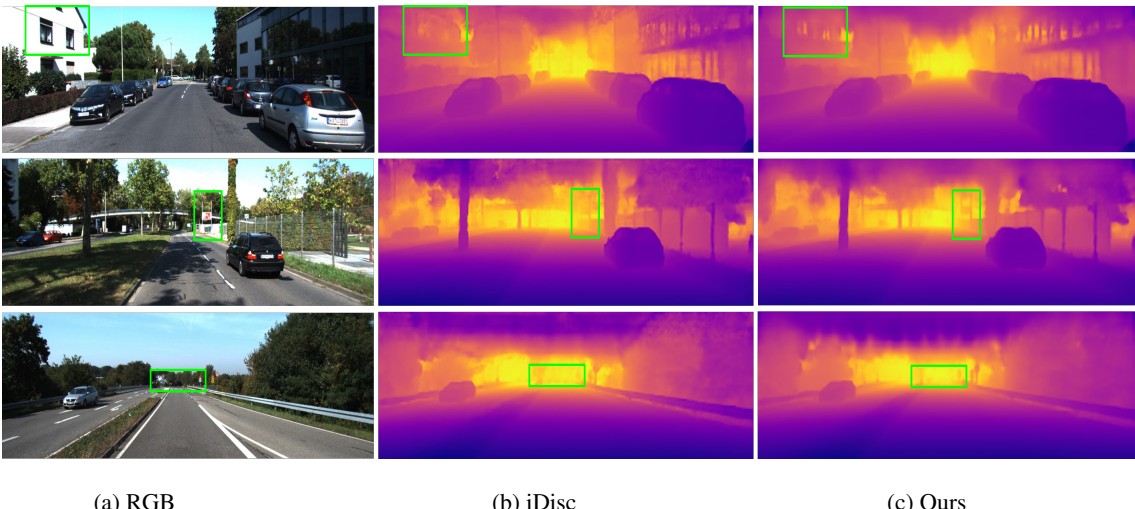

| (a) RGB | (b) iDisc | (c) Ours |
|---------|-----------|----------|

Figure 4: Qualitative comparison with the state-of-the-art on the KITTI dataset.

Table 3: Zero-shot testing of models trained on NYU. All methods are trained on NYU and tested without further fine-tuning on the official validation set of SUN-RGBD.

| Method | $\delta_1 \uparrow$ | Abs_REL $\downarrow$ | RMSE $\downarrow$ | $SI_{log} \downarrow$ |
|--------|---------------------|----------------------|-------------------|------------------------|
| BTS Lee et al. (2019) | 0.745 | 0.168 | 0.502 | 14.25 |
| AdaBins Bhat et al. (2021) | 0.768 | 0.155 | 0.476 | 13.20 |
| P3Depth Patil et al. (2022) | 0.698 | 0.178 | 0.541 | 15.02 |
| NeWCRFs Yuan et al. (2022) | 0.799 | 0.150 | 0.429 | 11.27 |
| iDisc Piccinelli et al. (2023) | **0.838** | **0.128** | **0.387** | 10.91 |
| Ours | 0.834 | **0.128** | 0.390 | **10.79** |

**NYU-V2.** See Table 1 for the comparison of the performance on the official NYU-Depth-v2 test set. Specifically, our results are comparable to the latest SOTA levels Piccinelli et al. (2023). The qualitative results in Figure 3 illustrate that our method estimates better depth especially in some difficult regions, such as plants in the house, pillows on the bed, dolls on the counter and cabinets and windows on the walls, etc. **KITTI.** Table 2 lists the performance metrics on the KITTI dataset. Our metric results are slightly worse than the latest SOTA,but the qualitative results in Figure 4 illustrate that our method gives better estimation of very distant objects. (e.g.,road traffic signs) and certain regular areas (e.g.,windows on the walls).

**SUN RGB-D.** To compare the generalisation performance,wu do a zero-shot testing by training our network on the NYU-Depth-v2 dataset and evaluate it on the test set of the SUN RGB-D dataset without any finetuning.Table 3 exhibits a compelling generalization performance.

## 5 CONCLUSION

In this work, we have presented a supervised monocular depth estimation network and achieved state-of-the-art results on the most popular datasets, NYU and KITTI. Benefiting from recent advances gated attention, we design an architecture block, called GABins, showing the capability of mining explicit relationship from internal feature maps for the desired prediction. In future work, we would like to investigate more deeply about relationship among different level features and their operations for depth estimation tasks.

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
