# OpenReview forum: "Gated Attention Bins for Depth Estimation"
_ICLR.cc/2024/Conference — Submitted to ICLR 2024_

### Official Review · Reviewer_H1hq · 2023-10-14

**Soundness:** 1 poor
**Presentation:** 2 fair
**Contribution:** 1 poor
**Rating:** 3
**Confidence:** 4

**Summary:**

Summary: This paper proposes a method called GABins for monocular depth estimation. The key ideas are:
1. Formulate depth prediction as a classification problem and adaptively discretize depth bins with gated attention.
2. Uses single-head gated attention along with Laplace normalization function to reduce computation and stabilize training.

**Strengths:**

1. The proposed method achieves SOTA result on two datasets.
2. The idea of make the depth bins adaptive is well-motivated.

**Weaknesses:**

The biggest weakness of this paper is that there is no evidence to support that the improvement in performance is a result of the proposed ideas. The authors only test GABins with Swin-Transformer, which is arguably one of the strongest backbones, and show improvements against other methods. It is therefore unclear whether this performance boost was achieved by adopting a better backbone, or due to the contribution of this paper. No ablation study was presented that could demonstrate the effectiveness of each components. All experiments are conducted with the full network. This can't convince me, or any other reader, that the proposed method is superior to other methods.

The experiment is also underwhelming. In 2023, it is not enough for a depth estimation paper to only test on NYUvs and KITTI. There are other larger datasets that can be used to better evaluate the depth estimation task.

As for the technical contribution itself, the main point proposed by the authors is the GABins, which is seems to be a small extension to an existing method (AdaBins). The only difference seems to be the adoption of gated attention, which is not a significant contribution in my view.

**Questions:**

N/A

---

### Official Review · Reviewer_Kve8 · 2023-10-29

**Soundness:** 1 poor
**Presentation:** 2 fair
**Contribution:** 1 poor
**Rating:** 1
**Confidence:** 1

**Summary:**

This paper proposes a method to solve the monocular depth estimation. To effectively leverage long-range features, they propose a  Gated Attention module. They transfer the depth regression problem to a classification problem and adjust the depth bins' width for each image. They achieve comparable performance with state-of-the-art methods on KITTI and NYU.

**Strengths:**

The paper is clearly present.

**Weaknesses:**

I cannot see any new insight from the paper. Most of the methods have been discussed or deeply studied by previous methods.
1. Converting the depth regression to a classification problem has been widely applied by previous methods. such as:
     a. Yuanzhouhan Cao, Zifeng Wu, Chunhua Shen, Estimating depth from monocular images as classification using deep fully convolutional residual networks, 2015;
     b. Wei Yin, Yifan Liu, Chunhua Shen, Youliang Yan, Enforcing geometric constraints of virtual normal for depth prediction
     c. Huan Fu, Mingming Gong, Chaohui Wang, Kayhan Batmanghelich, Dacheng Tao, Deep Ordinal Regression Network for Monocular Depth Estimation
     d. Shariq Farooq Bhat, Ibraheem Alhashim, Peter Wonka， AdaBins: Depth Estimation using Adaptive Bins
   Therefore, I cannot see any new insight and contributions in this part.

2. Attention modules have been widely applied to various vision tasks and network designs.
3. The paper compares with various state-of-the-art methods on NYU and KITTI benchmarks but loses basic ablations on the proposed modules.

**Questions:**

The paper proposes several attention modules, which one is the most effective one? The experiments do not include any ablations. I cannot confirm the effectiveness of the method.

---

### Official Review · Reviewer_vZWC · 2023-10-30

**Soundness:** 3 good
**Presentation:** 2 fair
**Contribution:** 2 fair
**Rating:** 3
**Confidence:** 3

**Summary:**

This work proposes a supervised monocular depth estimation network and test with public datasets on this task such as NYU and KITTI. Based on recent advances of gated attention, the authors propose an architecture block, named GABins, to improve the model performance. In particular, the proposed method formulizes monocular depth estimation as a classification problem by dividing the depth range into bins. The authors compute gated attention to obtain the association information between features to divide the depth range into bins and estimate their center values adaptively per image. The final depth values are estimated by linearly combining the results of the multiscale feature fusion and bin centers.

**Strengths:**

Monocular depth estimation is an important computer vision research topic and the authors address it with experiments on large scale real image datasets for this task. Comprehensive review is made about previous works. Experiments were performed with comparison against state of the art algorithms as baselines. The authors also performs zero-shot testing to validate the model’s capability of generalization.

**Weaknesses:**

Weakness of the paper:

(1) It is not novel to covert the depth estimation from regression to classification. It is ambiguous what are the real innovations of the proposed work compared to previous classification based method such as Fu et al. (2018) and Bhat et al. (2021, 2022).

(2) The improvement of accuracy compared to SOTA methods is marginal. As shown in Table 1 and Table 2, in both NYU-Depth-v2 DB and KITTI DB, the proposed method is weaker than iDisc.

(3) In the introduction section, two technical contributions are summarized, however, ablation study is missing to discuss the effectiveness of the different designed components.

There are a number of grammar and typo errors and the writing needs improvement.

**Questions:**

Questions for the authors:

(1) As shown in Table 1 and Table 2, in both NYU-Depth-v2 DB and KITTI DB, the proposed method is weaker than iDisc. Where is the claimed improvement?

(2) In the introduction section, two technical contributions are summarized. How does each of them contribute to the performance improvement?

---

### Official Review · Reviewer_C3QZ · 2023-11-01

**Soundness:** 2 fair
**Presentation:** 3 good
**Contribution:** 2 fair
**Rating:** 3
**Confidence:** 5

**Summary:**

This paper proposes the GABins module for accurate monocular depth estimation. The biggest drawback of this paper is insufficient experiments: This paper does not contain any ablation studies, which makes the statements less convincing. The novelty of the introduced method is not significant; several methods that adopt Swin-L backbones (VA-DepthNet, iDisc, IEBins) present similar or higher performance, crucially. If the authors believe that this method has an advantage in efficiency, it might be worth mentioning.

**Strengths:**

+ A module GABins to capture gated attention in the low-level features and divide the predicted depth range into bins where the bin widths change per image. The final depth values are estimated by linearly combining with the results of the multiscale feature fusion and bin centers.

+ A single-head gated attention unit other than multi-head self-attention to reduce computational complexity, and the Laplace normalization function to make the model train more stable.

**Weaknesses:**

I have the following major concerns with respect to the manuscript in its current form.  As I stated above, the weak and limited experiments make it hard to judge the contributions.

1) Study the contribution brought by classification and the GAU module separately. A baseline model can be: Swin transformer encoder with convolutional decoder and regression depth head. Then, add classification module with fixed discretization. Finally, add GAU module and perform classification with adaptive bins.

2) Compare model complexity in detail. This paper uses simple convolutional decoders, unlike a transformer decoder in NeWCRFs, which could be a weakness in terms of accuracy (compared to the ‘baseline’ mentioned in 1) ); thus, detailed metrics are important to address the significance of the contribution. Authors can compare model size, model complexity in detail. Authors also mentioned that the usage of ‘a single-head gated attention unit other than multi-head self-attention’ can reduce computational complexity; this should also be numerically compared.

3) Keep the backbone network fixed, and compare with other classification modules like AdaBins, LocalBins, BinsFormer, IEBins. There are so many similar ideas, and authors should address the significance and uniqueness of GAU.

[IEBins] IEBins: Iterative Elastic Bins for Monocular Depth Estimation

**Questions:**

Please address the issues as listed above in the weakness section.

-- The constructed network or the highlighted module GAU is difficult to be proved effective due to lack of ablation study.

-- In Figuire 3,the depth map of iDsc at the third row does not match the original image.

-- The depth map prediction results of KITTI and NYU presented in this article do not have significant advantages,which may not demonstrate the effectiveness of the proposed method.

---

### Meta-Review · Area_Chair_uTpE · 2023-11-30

**Metareview:**

Four expert reviewers recommended rejection and no rebuttal was submitted. The AC found no reason to overturn the reviewers' recommendation for rejection.

**Justification For Why Not Higher Score:**

Four expert reviewers recommended rejection and no rebuttal was submitted.

**Justification For Why Not Lower Score:**

NA

---

### Decision · Program_Chairs · 2024-01-16

Reject